evolution, genomics, plant science

admixture, C$_4$ photosynthesis, miombo woodlands, phylogenomics, phylogeography, polyploidy

**Author for correspondence:**
Pascal-Antoine Christin
e-mail: p.christin@sheffield.ac.uk

†Present address: Institut Botànic de Barcelona (IBB, CSIC-Ajuntament de Barcelona), Passeig del Migdia s.n., 08038 Barcelona, Catalonia, Spain.
‡Lancaster Environment Centre, Lancaster University, Lancaster LA1 4YQ, UK.
¶Section for GeoGenetics, GLOBE Institute, University of Copenhagen, Øster Farimagsgade 5, building 7, 1353 København K, Denmark.

# Contrasted histories of organelle and nuclear genomes underlying physiological diversification in a grass species

Matheus E. Bianconi[1], Luke T. Dunning[1], Emma V. Curran[1], Oriane Hidalgo[2,†], Robyn F. Powell[2], Sahr Mian[2], Ilia J. Leitch[2], Marjorie R. Lundgren[1,‡], Sophie Manzi[3], Maria S. Vorontsova[2], Guillaume Besnard[3], Colin P. Osborne[1], Jill K. Olofsson[1,¶] and Pascal-Antoine Christin[1]

[1]Department of Animal and Plant Sciences, University of Sheffield, Western Bank, Sheffield S10 2TN, UK
[2]Comparative Plant and Fungal Biology, Royal Botanic Gardens, Kew, Richmond, Surrey TW9 3AB, UK
[3]Laboratoire Evolution and Diversité Biologique (EDB UMR5174), Université de Toulouse III – Paul Sabatier, CNRS, IRD, 118 route de Narbonne, 31062 Toulouse, France

 MEB, 0000-0002-1585-5947; IJL, 0000-0002-3837-8186; CPO, 0000-0002-7423-3718;
P-AC, 0000-0001-6292-8734

C$_4$ photosynthesis evolved multiple times independently in angiosperms, but most origins are relatively old so that the early events linked to photosynthetic diversification are blurred. The grass *Alloteropsis semialata* is an exception, as this species encompasses C$_4$ and non-C$_4$ populations. Using phylogenomics and population genomics, we infer the history of dispersal and secondary gene flow before, during and after photosynthetic divergence in *A. semialata*. We further analyse the genome composition of individuals with varied ploidy levels to establish the origins of polyploids in this species. Detailed organelle phylogenies indicate limited seed dispersal within the mountainous region of origin and the emergence of a C$_4$ lineage after dispersal to warmer areas of lower elevation. Nuclear genome analyses highlight repeated secondary gene flow. In particular, the nuclear genome associated with the C$_4$ phenotype was swept into a distantly related maternal lineage probably via unidirectional pollen flow. Multiple intraspecific allopolyploidy events mediated additional secondary genetic exchanges between photosynthetic types. Overall, our results show that limited dispersal and isolation allowed lineage divergence, with photosynthetic innovation happening after migration to new environments, and pollen-mediated gene flow led to the rapid spread of the derived C$_4$ physiology away from its region of origin.

## 1. Introduction

Most terrestrial plants assimilate carbon using the ancestral C$_3$ photosynthetic metabolism, but the efficiency of this pathway decreases in conditions that limit CO$_2$ availability within the leaf, such as warm, arid and saline habitats [1]. The derived C$_4$ metabolism boosts productivity in such conditions via the concerted action of numerous enzymes and anatomical features that concentrate CO$_2$ at the site of photosynthetic carbon fixation [1]. Nowadays, C$_4$ plants, particularly those belonging to the grass and sedge families, dominate tropical grasslands and savannahs, which they have shaped via feedback with herbivores and fire [2,3]. The C$_4$ metabolism evolved multiple times independently over the past 30 Myr [1], and retracing the eco-evolutionary dynamics linked to photosynthetic transitions is difficult for old C$_4$ lineages. However, a few lineages evolved the C$_4$ trait relatively recently, offering tractable systems to study the events leading to C$_4$ evolution.

The grass *Alloteropsis semialata* is the only species known to have genotypes with distinct photosynthetic pathways [4]. $C_4$ accessions are distributed across the palaeotropics, while $C_3$ individuals are restricted to southern Africa (electronic supplementary material, figure S1 [5]). In addition, individuals performing a weak $C_4$ pathway ($C_3 + C_4$ individuals [6]) occur in parts of Tanzania and Zambia [5], in the plant biogeographic region referred to as 'Zambezian' [7] or 'Central Zambezian' [8], which is associated with miombo woodlands [9]. Analyses of plastid genomes have suggested that the species originated in this region, and one lineage associated with the $C_3$ type then migrated to southern Africa, while a $C_4$ lineage dispersed across Africa and Asia/Oceania [10]. However, these previous phylogenetic trees had limited resolution, and more data are needed to firmly establish the exact origin of the different lineages.

The different photosynthetic types of *A. semialata* are associated with distinct ploidy levels in South Africa, but both $C_4$ and non-$C_4$ diploids exist in other parts of Africa [4,10], and nuclear genome analyses have found evidence of genetic exchanges between lineages with different photosynthetic types [11]. In addition, previously reported discrepancies between mitochondrial and plastid genomes [12] might reflect the footprint of intraspecific allopolyploidization, as previously suggested based on cytological analyses [13]. However, the history of nuclear exchanges and their effect on the spread of different photosynthetic types through ecological and geographical spaces remain to be formally established.

In this study, we analyse the organelle and nuclear genomes of 69 accessions of *A. semialata* (plus six congeners) from 28 countries, covering the known species range and photosynthetic diversity, to establish the order of seed-mediated range expansion and subsequent pollen-mediated admixture of nuclear genomes. Organelle phylogenetic trees are used to (i) identify the geographical and ecological origins of the species and its subgroups, with a special focus on the $C_4$ group. Analyses of nuclear genomes are then used to (ii) establish the history of secondary genetic exchanges and their impact on the sorting of photosynthetic types. Finally, genome size estimates coupled with phylogenomics and population genomics approaches are used to (iii) identify the origins of polyploids and their relationship to photosynthetic divergence. Our detailed genome biogeography analyses shed new light on the historical factors that lead to functional diversity within a single species.

## 2. Materials and methods

### (a) Sampling, sequencing and data filtering

Whole-genome sequencing data for 26 accessions of *A. semialata* and one of *A. paniculata* were sequenced here and added to 48 accessions retrieved from previous studies (electronic supplementary material, table S1). For herbarium samples, genomic DNA (gDNA) was isolated using the BioSprint 15 DNA Plant Kit (Qiagen). Libraries were prepared with 22–157 ng of gDNA using the Illumina TruSeq Nano DNA LT Sample Prep kit (Illumina, San Diego, CA, USA). Each sample was sequenced at the GenoToul-GeT-PlaGE platform (Toulouse, France) as paired-end reads on 1/24th of an Illumina HiSeq3000 lane. For fresh or silica gel dried leaves, gDNA was isolated using the Plant DNeasy Extraction kit (Qiagen). Libraries were constructed by the respective sequencing centres and paired-end sequenced on full, 1/6th or 1/12th lane of an Illumina HiSeq2500 at the Sheffield Diagnostic Genetics Service (UK) and the Edinburgh Genomics facility (UK; electronic supplementary material, table S1). The expected sequencing depth ranged from 0.6 to 58.7 × (median = 4.9×; electronic supplementary material, table S1). Raw Illumina datasets were filtered before analysis using the NGSQC Toolkit v. 2.3.3 [14] to remove reads with less than 80% of the bases with Phred score above 20, reads containing ambiguous bases or adaptor contamination. The retained reads were further trimmed from the 3′ end to remove bases with Phred score below 20. The quality of the filtered datasets was assessed using FastQC v. 0.11.9 [15].

### (b) Genome sizing and carbon isotope analyses

Genome sizes of *A. semialata* accessions were retrieved from previous studies or estimated by flow cytometry (electronic supplementary material, table S1) following the one-step protocol of [16] with minor modifications [17]. For fresh samples either the Ebihara or GPB with 3% PVP nuclei isolation buffer was used, whereas the CyStain PI Oxprotect buffer (Sysmex, Germany) was used for silica-dried material. As internal calibration standards, *Petroselinum crispum* 'Champion Moss Curled' (4.5 pg/2C) or *Oryza sativa* IR36 (1.0 pg/2C) were used for diploids, while *Pisum sativum* 'Ctirad' (9.09 pg/2C) was used for accessions with C-values more than three times larger than any diploid. All samples were analysed on a Sysmex Partec Cyflow SL3 flow cytometer fitted with a 100 mW green solid-state laser (Cobalt Samba, Sweden). Individuals with known chromosome numbers and genome sizes [10] were used to assign ploidy levels based on genome size estimates.

Photosynthetic types were established based on carbon isotope ratios retrieved from previous studies or measured here as previously described [10] (electronic supplementary material, table S1). All individuals with values below −17‰ were classified as non-$C_4$. These were further distinguished between $C_3$ and $C_3 + C_4$ using previous anatomical and physiological data, and/or expression levels of $C_4$ pathway genes, where available (electronic supplementary material, table S1).

### (c) Assembly of organelle genomes and molecular dating

Plastid and mitochondrial genome sequences were assembled here using a reference-guided approach, except for 15 plastid genomes retrieved from previous studies (electronic supplementary material, table S1). The reference dataset consisted of the chromosome-level nuclear, mitochondrial and plastid genomes of an Australian individual of *A. semialata* (AUS1-01 [10,18]). Paired-end genomic reads were mapped to this reference using Bowtie2 v. 2.3.5 [19] with default parameters. Variant sites from the reads uniquely mapped to each organelle were incorporated into a majority consensus sequence using the mpileup function of Samtools v. 1.9 [20] implemented in a bash-scripted pipeline [12]. Only sites covered by more than five times the expected sequencing depth of the nuclear genome (electronic supplementary material, table S1) were called, discarding potential organelle-nuclear transfers. This approach produced sequences that were already aligned to the organelle references. The plastid alignment was manually combined with the 15 previous sequences using Aliview v. 1.17.1 [21], after the latter were aligned using MAFFT v. 7.427 [22], and the second inverted repeat was removed. Both organelle alignments were then trimmed to remove sites covered by less than 90% of individuals using trimAl v. 1.4 [23]. The resulting plastid and mitochondrial alignment lengths were 84 587 bp and 139 916 bp, respectively (available on Dryad: https://dx.doi.org/10.5061/dryad.zs7h44j6v [24]).

A time-calibrated phylogeny was obtained independently on plastid and mitochondrial alignments using BEAST v. 1.8.4 [25].

The median ages estimated in [10] were used for secondary calibration of the genus *Alloteropsis* (11.46 Ma for the crown node, and 8.075 Ma for the split between *A. angusta* and *A. semialata*), using a normal distribution with standard deviation of 0.0001. While secondary calibrations are imperfect and can result in younger estimates and underestimated uncertainties [26], they are the only option in the absence of fossils of *Alloteropsis* and provide accurate relative ages and indicative absolute ages. The GTR + G + I substitution model was used, with a lognormal uncorrelated relaxed clock and a constant population size coalescent tree prior. Two analyses were run in parallel for 300 000 000 generations using the CIPRES Science Gateway 3.3, with sampling every 20 000 generations. Convergence of runs and effective sample sizes greater than 100 were confirmed using Tracer v. 1.6 [27]. Median ages of trees sampled after a burn-in period of 10% (plastid) and 25% (mitochondria) were mapped on the maximum clade credibility tree.

## (d) Phylogenetic analyses of the nuclear genome

Genome-wide nuclear markers were assembled using the genomic data and combined into a multigene coalescent phylogeny, which can identify different histories among genes. A total of 7408 single-copy orthologues of Panicoideae were identified from the genomes of *A. semialata* [18], *Setaria italica*, *Panicum hallii* and *Sorghum bicolor* (from Phytozome v. 13 [28]) using OrthoFinder v. 2.3.3 [29]. Coding sequences of *A. semialata* were extracted and used as references to assemble genes from all *Alloteropsis* accessions with the approach described above for organelle genomes, except that reads were mapped as unpaired to avoid discordant pairs where mates mapped to non-exonic sequences. Sites covered by less than 70% of individuals were trimmed using trimAl, and individual sequences shorter than 200 bp after trimming were discarded. Only trimmed alignments longer than 500 bp and with taxon occupancy greater than 95% were retained. A maximum likelihood tree was then inferred on each of the 3553 retained alignments using RAxML v. 8.2.4 [30], with a GTR + CAT substitution model and 100 bootstrap pseudoreplicates. Gene trees were summarized into a multigene coalescent phylogeny using Astral v. 5.6.2 [31] after collapsing branches with bootstrap support values below 30. The analysis was repeated with only confirmed diploid individuals, and with all individuals but less stringent missing data thresholds; (i) non-trimmed alignments, and (ii) alignments trimmed to remove sites covered by less than 30% of individuals.

Using a similar approach to [32], we evaluated the probability of observing the organelle topology solely based on incomplete lineage sorting. A total of 100 000 gene trees were simulated in Hybrid-Lambda [33], using the nuclear coalescent phylogeny for diploid accessions, with terminal branches assigned an arbitrary length of one and extended to make the tree ultrametric, and branch lengths multiplied by two to reflect the smaller effective population size of organelles in monoecious species [34]. Simulations were performed using default parameters and repeated using various combinations of mutation rates (from $2.5 \times 10^{-5}$ to $10^{-4}$) and population sizes (from 100 to 50 000).

## (e) Genetic structure

Principal component and individual-based admixture analyses were performed on reads mapped to the whole nuclear genome. Reads were sorted and indexed using Samtools v. 1.9, and duplicates were removed using the function MarkDuplicates from Picard tools v. 2.13.2 (http://broadinstitute.github.io/picard/). Genotype likelihoods were estimated using ANGSD v. 0.929 [35]. Sites covered by 70% or more of individuals and with mapping and base quality scores of 30 or above were retained, resulting in 11 439 variable sites. A covariance matrix was estimated from the genotype likelihoods using PCAngsd

v. 0.98 [36]. Eigenvector decomposition was carried out using the *eigen* function in R v. 3.4.4 to recover the principal components of genetic variation. Individual admixture proportions were estimated from genotype likelihoods using a maximum likelihood approach with NGSadmix v. 32 [37]. NGSadmix was run with numbers of ancestral populations ($K$) ranging from 1 to 10, with five replicates each and random starting seeds. The Evanno method [38], as implemented in CLUMPAK [39], identified the value of $K$ that best describes the uppermost level of structure.

## (f) Genome composition

The multigene coalescence approach implemented above allows for different histories of loci, but not for different histories of alleles at each locus, as expected in diploid hybrids or allopolyploids. Likewise, the population genomics tools used above are not tailored for mixed ploidy datasets [40], and do not explicitly consider known ploidy differences. We consequently established the history of polyploids with phylogenetic trees of phased alleles obtained with a newly developed pipeline. To reduce risks of paralogy, we considered only near-universally single-copy orthologues in land plants according to BUSCO v. 3.0.2 [41]. Out of 1202 such genes identified in *A. semialata*, 473 had at least one exon longer than the average insert size of the reads used here. The longest exon of each of these genes was used. As a first step to verify that the genes were single-copy and orthologous, we considered only the 26 individuals of *A. semialata* that were diploid, including a F1 hybrid between $C_3$ (nuclear clade I) and $C_4$ (nuclear clade IV) parents (electronic supplementary material, table S1). These individuals were sequenced as 250 bp paired-end reads (insert size = 550 bp) with an expected sequencing depth between 4.7 and 58.8, and we added four congeners (three *A. angusta* and one *A. cimicina*) sequenced at similar depth to serve as outgroups. Reads were mapped to the reference genome of *A. semialata* using Bowtie2 with default parameters, except the insert size (-X) that was increased to 1100 bp. Mapped reads were then phased for the 473 exons using the phase function of Samtools v. 0.1.19, where bases with quality below 20 were removed during heterozygous calling (option -Q20), and reads with ambiguous phasing were discarded (option -A). Sequences were then generated for both alleles using custom bash scripts in which different depth filters were applied for resequencing and high coverage (20× or above) datasets (at least 3 and 10 reads covering each position and at least 2 and 3 reads covering each variant for polymorphic sites, respectively). Phased sequences shorter than 200 bp were discarded. We only retained genes that: (i) included at least one sequence for each of the two congeners; (ii) had at least four sequences in each of the four nuclear clades of *A. semialata* (see Results); and (iii) had at least 30 sequences in total (50% of the possible maximum). A maximum likelihood tree was inferred for each of the 300 genes that matched these criteria using PhyML v. 20120412 [42] with a GTR + G + I model. The resulting trees were rooted on *A. cimicina* and processed with custom scripts to identify those genes for which one allele of the F1 hybrid was nested within the $C_3$ clade I and one nested within the $C_4$ clade IV. The 120 genes not fulfilling this criterion were discarded, as they might represent insufficiently variable genes or include *Alloteropsis*-specific paralogs. The remaining 180 phylogenetically informative genes were retained for downstream analysis.

Because phasing is difficult with polyploids, we incorporated each pair of reads from the polyploids into the phylogenetic tree containing the diploid phased alleles and assigned them to the clade in which they were positioned. These analyses were conducted on five hexa- and dodecaploids of *A. semialata* sequenced as 250-bp paired-end reads (electronic supplementary material,

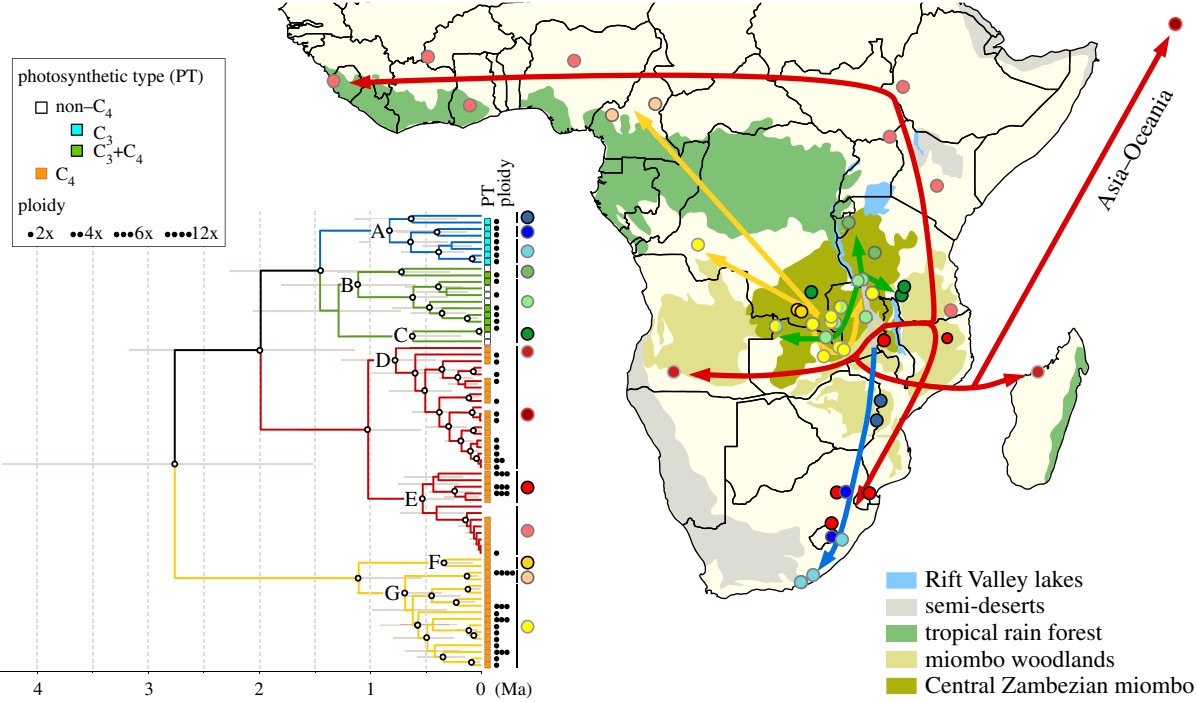

**Figure 1.** Origin and dispersal of *Alloteropsis semialata* in Africa. The time-calibrated phylogenetic tree based on mitochondrial genomes is shown, with letters on nodes (A–G) indicating the organelle lineages (see electronic supplementary material, figure S2 for details). White dots indicate nodes with posterior probabilities of 0.95 or above and grey bars represent 95% HPD intervals around estimated ages. For each sample, the photosynthetic type is indicated with a coloured square and the ploidy level by the number of black dots. All sampled African populations are shown on the map, with circles coloured based on the group indicated on the right of the phylogeny. Arrows indicate putative dispersal events. (Online version in colour.)

table S1). Each pair of reads that fully overlapped with one of the 180 exons, as determined using BEDTools v. 2.24.0 [43] with the function 'intersect' (option -f 1.0), was separately added to the respective gene alignment using the MAFFT function '–add'. The paired reads were then merged, and each alignment was trimmed to remove non-overlapping regions (max. alignment length = 500 bp). Individual sequences shorter than 250 bp were subsequently discarded. A maximum likelihood tree was then inferred as described above, and the read pair was assigned to the nuclear clade in which it was nested. Read assignments were not used in cases in which the $C_3$ and $C_4$ alleles from the F1 hybrid were not correctly placed as a consequence of alignment trimming, or when the sister group of the reads was composed of multiple lineages. The analyses were later repeated with each diploid used as the focus individual, in which case the phased alleles from the focal individual were removed from the reference dataset.

## 3. Results

### (a) Genome sizes

Out of 38 samples with genome sizes available, 31 were within 25% of the range previously reported for *A. semialata* diploids (1.78–2.77 pg/2C; electronic supplementary material, table S1). One individual from Australia is possibly a tetraploid (3.65 pg/ 2C), while three from Zambia and one from Mozambique have genome sizes suggesting hexaploidy (5.35 and 6.71 pg/2C), as previously reported for some South African populations [10,13]. Finally, individuals from one Cameroonian population had genome sizes suggesting dodecaploids (11.87 pg/2C; electronic supplementary material, table S1). All polyploids detected so far in *A. semialata* have carbon isotopic signatures of $C_4$ plants (electronic supplementary material, table S1).

### (b) Time-calibrated organelle phylogenies

The organelle phylogenetic trees recovered the seven major lineages reported in previous studies, as well as a clear incongruence between the two organelles (figure 1 and electronic supplementary material, figure S2 [10–12]). Indeed, six individuals, including one hexaploid and one dodecaploid, from Cameroon, Democratic Republic of Congo (DRC) and Zambia form a monophyletic group within plastid lineage DE, but form a paraphyletic group within mitochondrial lineage FG (electronic supplementary material, figure S2).

The ages estimated on the mitochondrial and plastid genomes were overall similar, and discrepancies can result from the use of secondary calibrations. The first split within *A. semialata*, which separates lineage FG from the Central Zambezian region of Africa (in DRC, Zambia and Tanzania) and the rest of the species (lineage ABCDE), was estimated at 2.8/1.8 Ma on the mitochondrial/plastid trees (95% HPD = 1.5–4.3/1.2–2.6). Within the FG group, accessions from DRC diverge first, and accessions from the south of Tanzania are nested within a paraphyletic Zambian clade (figure 1 and electronic supplementary material, figure S2). Within the ABCDE group, the separation between the non-$C_4$ group ABC and the exclusively $C_4$ lineage DE is estimated at 2/1.7 Ma. Accessions from the B and C clades are spread across northern areas of the Central Zambezian region (Burundi, DRC, and Tanzania), with Zambian accessions derived from within part of clade B (figure 1 and electronic supplementary material, figure S2). Within clade A covering southern Africa, early divergence from accessions from Mozambique and Zimbabwe likely represents the footprint of a gradual migration to South Africa between 1.4 and 0.3 Ma (figure 1 and electronic supplementary material, figure S2).

Proc. R. Soc. B 287: 20201960

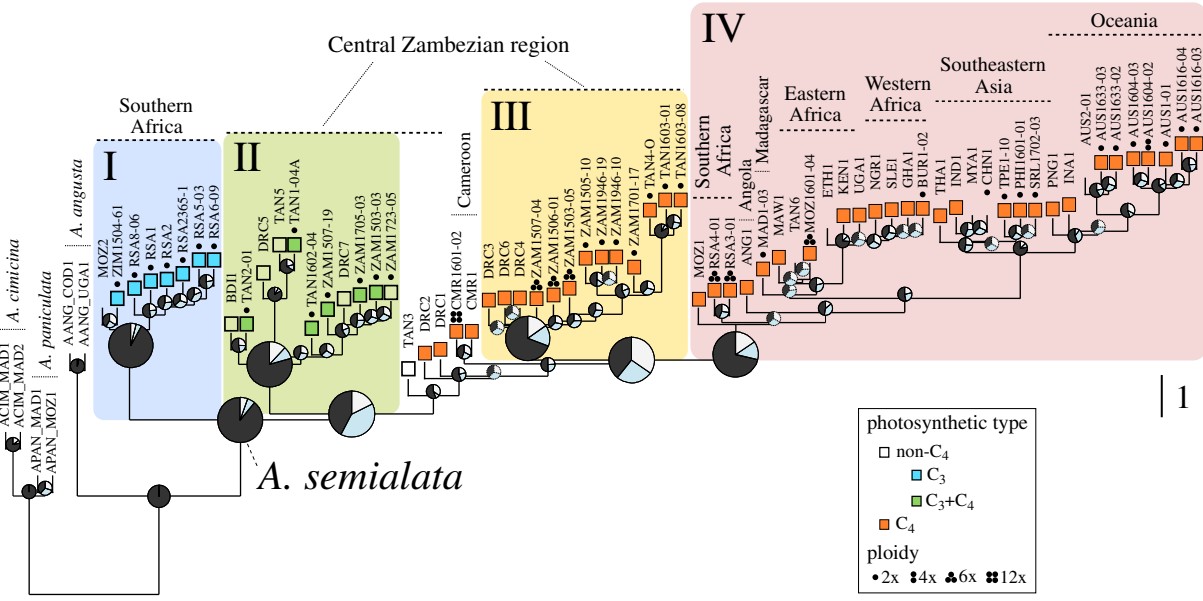

**Figure 2.** Nuclear history of *Alloteropsis*. The multigene coalescent species tree was estimated from 3553 genome-wide nuclear markers. Pie charts, magnified for key nodes, indicate the proportion of quartet trees that support the main (dark grey), first (pale blue) and second (light grey) alternative topologies. Dashed-line pie charts indicate nodes with local posterior probability below 0.95. Branch lengths are in coalescent units, except the terminal branches, which are arbitrary. Roman numbers I–IV denote the four main nuclear clades of *A. semialata*, which are indicated with coloured shades. Major geographical regions are indicated. (Online version in colour.)

Within the $C_4$ organelle lineage D, accessions from Asia, Oceania and Madagascar are sister to a sample from Angola (figure 1 and electronic supplementary material, figure S2). The sister lineage E contains a subgroup spread east of the Central Zambezian region (Tanzania, Malawi and Mozambique) and South Africa, while the other subgroup contains one accession from Ethiopia that is sister to samples spread from Kenya to Sierra Leone with very little divergence (figure 1 and electronic supplementary material, figure S2). The six individuals with discordant mitochondria and plastids are placed as sister to this clade E in the plastid phylogeny.

## (c) Nuclear phylogeny

A multigene coalescent phylogeny was estimated based on 3553 nuclear markers. The four nuclear clades previously defined within *A. semialata* [11] were retrieved with high support (figure 2), and similar relationships were obtained with different thresholds for missing data (electronic supplementary material, figure S3), and when only diploid samples were included (electronic supplementary material, figure S4). The monophyly of nuclear clade I, which corresponds to organelle lineage A and is associated with $C_3$ photosynthesis, is supported here by almost all quartet trees. A lower proportion of quartet trees (~75%) support the monophyly of nuclear clade II, which contains non-$C_4$ accessions from the Central Zambezian region ($C_3 + C_4$; organelle lineages B and C). This proportion is even lower for each of the nuclear clades III and IV (~66%), which contain the $C_4$ accessions (figure 2). However, the two alternative topologies occur at similar frequencies at the base of each of the $C_4$ clades (figure 2), which is compatible with incomplete lineage sorting. The relationships among the four clades vary, with the $C_3 + C_4$ clade II placed either as sister to clade III + IV (figure 2) or to clade I (electronic supplementary material, figures S3 and S4), in both cases with a similar number of quartets supporting the alternative topology.

Five accessions are 'unplaced' at the base of $C_4$ nuclear clades III plus IV with low quartet support values (figure 2 and electronic supplementary material, figure S3). These include two $C_4$ Cameroonian accessions (one of which is a dodecaploid) and another three accessions (two $C_4$ and one non-$C_4$) from DRC and Tanzania, which are admixed between clades II and III [11]. The exclusively $C_4$ nuclear clade III is formed of accessions from mitochondrial lineage FG (including those placed within plastid lineage E) that are restricted to the Central Zambezian region. Nuclear clade IV contains all $C_4$ individuals from mitochondrial lineage DE, but with relationships that differ from the organelle genomes. The first splits within clade IV lead to South African hexaploids, one Mozambican hexaploid accession and one Angolan accession, while most accessions cluster in one of two sister groups; one composed of all other African accessions (including Madagascar) and one composed exclusively of Asian and Oceanian accessions (figure 2 and electronic supplementary material, figures S3 and S4).

In our simulations of gene trees based on the nuclear coalescent phylogeny, only 5% of trees where the four nuclear clades were monophyletic mirrored the organellar topology (i.e. clade III sister to clades I + II + IV). Similar results were obtained across the range of parameter values explored here (coefficient of variation = 0.3–1.1%).

## (d) Population structure and genome composition

A principal component analysis grouped individuals largely according to their nuclear phylogenetic relationships (electronic supplementary material, figure S5). Admixture analyses identified four and seven clusters as good fits for the data, and again retrieved groups that match the nuclear phylogeny (electronic supplementary material, figure S6). The five unplaced individuals were positioned in between clades in the principal component analysis (electronic supplementary material, figure S5) and showed mixed ancestry (electronic supplementary material figure S6).

*Proc. R. Soc. B* **287**: 20201960

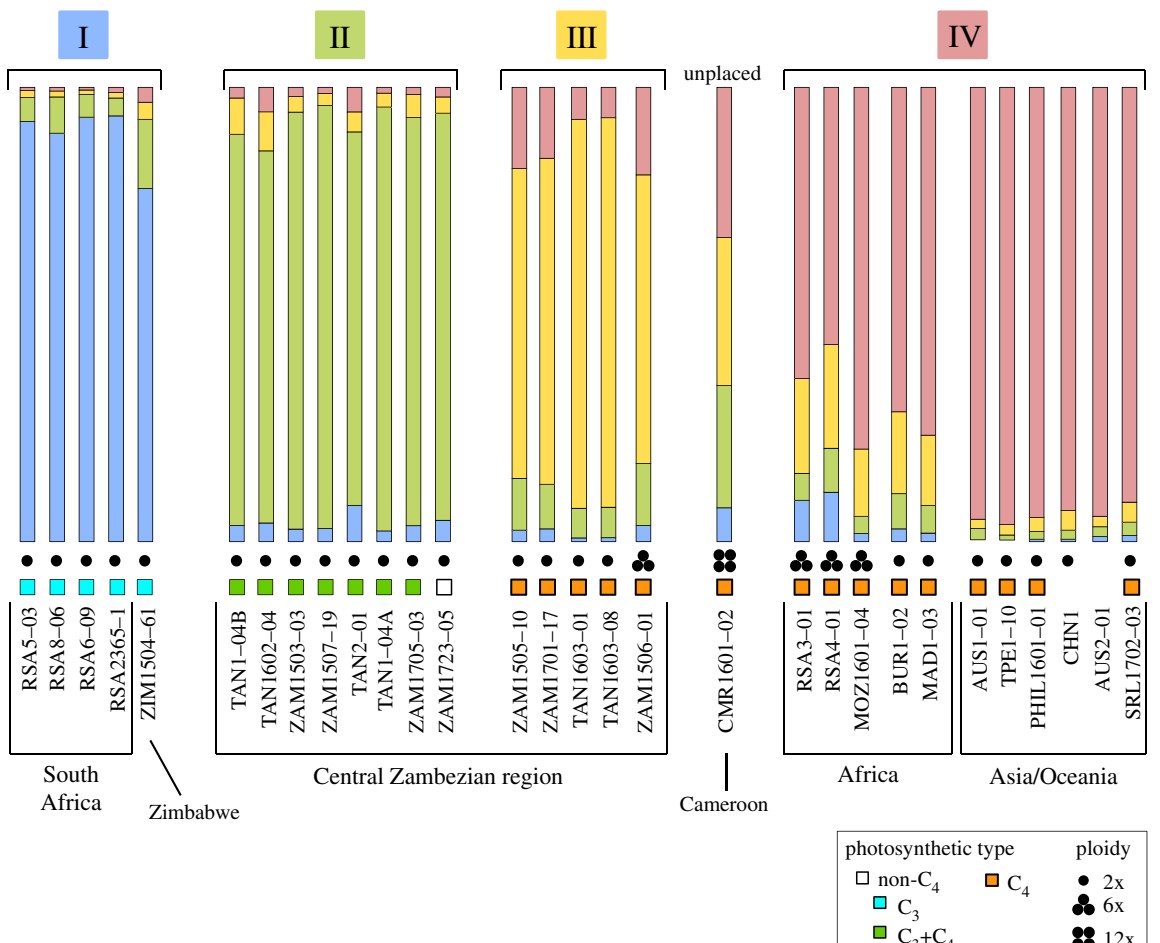

**Figure 3.** Genomic composition of *Alloteropsis semialata*. The proportion of alleles of single-copy exons assigned to each of the four nuclear clades of *A. semialata*, numbered and coloured as in figure 2, are represented by colours in bars (see electronic supplementary material, table S2). (Online version in colour.)

The genome composition analysis showed that the vast majority of reads from the Asian/Oceanian individuals (more than 90%) were assigned to the $C_4$ nuclear clade IV (figure 3; electronic supplementary material, table S2), confirming the expectations based on the multigene coalescent phylogeny. Low levels of assignment to other nuclear clades might represent incomplete lineage sorting or methodological noise. Similar high levels of assignments to the expected group were observed in most individuals from $C_3$ clade I and $C_3 + C_4$ clade II, but this number dropped to 82% in some individuals from clade II and to 78% in the Zimbabwean sample from clade I (figure 3; electronic supplementary material, table S2). The proportion of reads of $C_4$ diploids from Africa assigned to the expected clade (either III and IV) was as low as 68%, with up to 18% and 11% of reads assigned to the other $C_4$ clade and to the non-$C_4$ clade II, respectively. The ancestry of the polyploid individuals differed between geographical regions. In particular, the reads from the dodecaploid individual from Cameroon were almost equally spread among the $C_4$ clades III and IV and the non-$C_4$ clade II (figure 3; electronic supplementary material, table S2).

## 4. Discussion

### (a) Limited seed dispersal in the region of origin

The organelle genomes, which are mostly maternally inherited, track the history of seed dispersal in plants. In *A. semialata*, four organelle lineages capturing the earliest splits within the species (B, C, F and G) are restricted to the Central Zambezian miombo woodlands dominated by *Brachystegia* and *Julbernardia* trees (figure 1 and electronic supplementary material, figure S2) [9]. *Brachystegia* was already present in eastern Africa during the late Oligocene [44], and throughout the Miocene [45], thus the first divergence within *A. semialata* (~2–3 Ma) likely happened in this biome. The extent of miombo woodlands varied with glaciation cycles [46,47], and restrictions to dispersal during glacial maxima might have driven the vicariance of organelle lineages FG and ABCDE in the Pleistocene, as previously reported for other taxa occurring across the Great Rift System [48]. Indeed, the order of splits within group FG is compatible with an origin west of the Rift Valley lakes, while the well-supported relationships within lineages B and C support their origin to the east of the lakes (figure 1). The present-day co-occurrence of FG and BC organelle groups probably follows a migration beyond their refugia after the re-expansion of miombo woodlands (figure 1) [47,49,50].

Despite having originated about 2 Ma, lineages FG and BC still occur within a relatively small geographical region in central/eastern Africa. The visible geographical structure of each of these lineages in the organelle phylogenies (figure 1) further supports limited seed dispersal. These two lineages occur in the wet miombo that occupies the mountains separating the Zambezi and Congo basins. Variations in elevation coupled with relatively dense tree cover might limit seed dispersal for this species with seeds spread mainly by gravity. By contrast, the lineages that escaped this centre of origin

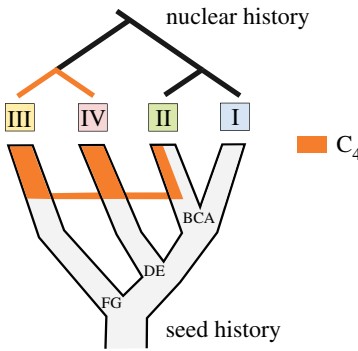

**Figure 4.** Putative history of the $C_4$ nuclear genome of *Alloteropsis semialata*. The $C_4$ nuclear genome is shown in orange, on top of the seed history. Gene flow between lineages is indicated by horizontal connections. (Online version in colour.)

bear the footprint of a rapid geographical spread. Over the last million years, lineage A migrated to the south of Africa, and the ancestor of lineage DE reached the lowland surrounding the wet Central Zambezian miombo from where it rapidly spread around the world (figure 1). The rapid migration of lineage DE was facilitated by the broader niche conferred by its $C_4$ photosynthetic type [10], but the concurrent dispersal of $C_3$ lineage A suggests that corridors of low elevation east, north and south of the Central Zambezian region, coupled with open grasslands and savannahs in these regions, facilitated the long-distance spread of *A. semialata* seeds outside the Central Zambezian region.

## (b) Widespread pollen flow and sweep of the $C_4$ nuclear genome

The organelle lineages are loosely associated with distinct nuclear groups (figures 1 and 2), indicating that the split of seed-transported organelle lineages was accompanied by a reduction of nuclear exchanges. However, the nuclear structure is less marked and numerous discrepancies between nuclear and organelle phylogenies indicate secondary genetic exchanges mediated by pollen. Such cytoplasmic-nuclear discordances are widespread in plants and animals [51,52], and have revealed complex patterns of lineage diversification [53–55].

The organelle phylogenies consistently identify two distinct $C_4$ groups (FG and DE; figures 1, 2 and electronic supplementary material, figure S2), while all $C_4$ accessions are monophyletic in nuclear analyses (figure 2 [11,18]). Our simulations show that such patterns are unlikely to result solely from incomplete lineage sorting. Instead, the sister group relationship between nuclear clades III and IV, which are associated with divergent organelles, suggests the swamping of one nuclear genome lineage by the other (figure 4). The directionality of this exchange is unknown, but repeated, unidirectional gene flow mediated by pollen must have occurred in a region where only monoparentally inherited organelles persisted [56], as previously reported for other taxa [57]. One of these organelle lineages originates from the Central Zambezian highlands (lineage FG), while the other originates from the lowlands of eastern Africa (lineage DE; figure 1). Differences in elevation along with predominantly easterly winds could have restricted organelle transport via seed migration but favoured nuclear gene flow via pollen movements from the lowlands to the west. Given that $C_4$ genes are encoded by the nuclear genome, it is tempting to

hypothesize that an efficient $C_4$ trait evolved following migration to lower elevation, where higher temperatures increased selective pressures for photosynthetic transitions. Pollen flow would then have brought the $C_4$ pathway to populations from the highlands, where the sweep would have been mediated by selection for the derived pathway with its broader ecological niche [10,58]. The marked incongruence between organelle and nuclear phylogenies thus indicates that nuclear genes encoding the $C_4$ trait were rapidly spread to other habitats by hijacking seeds of the same species.

## (c) Recurrent hybridization and polyploidization

The comparison of nuclear genomes suggests episodic hybridization between the different lineages. Similar proportions of quartet trees place the $C_3 + C_4$ clade II as sister to the $C_3$ clade I and the $C_4$ clades III + IV (figure 2 and electronic supplementary material, figures S3 and S4), which is not predicted with only incomplete lineage sorting [59]. Instead, the patterns point to an ancient episode of hybridization that might have brought some genes adapted for the $C_4$ pathway into a different genomic background (figure 4). Indeed, some genes for $C_4$ enzymes group $C_3 + C_4$ and $C_4$ individuals [6,11], highlighting the importance of hybridization for photosynthetic diversification [60]. After this ancient introgression, the $C_4$ and $C_3 + C_4$ types evolved mostly independently despite their close geographical proximity, but African $C_4$ from both clades III and IV possess alleles that group with $C_3 + C_4$ individuals (figure 3), pointing to recurrent gene flow.

Besides polyploids occurring in South Africa [4,10,13], we report here, for the first time, hexaploids from Zambia and Mozambique, and a dodecaploid from Cameroon (electronic supplementary material, table S1). The genomic compositions differ among the polyploids, which are placed in different parts of the organelle phylogenies (figure 3 and electronic supplementary material, figure S2). These patterns suggest a minimum of three independent polyploidization events; (i) admixture between $C_4$ nuclear clades III and IV leading to hexaploids from Mozambique and South Africa, (ii) contributions of nuclear clades II, III and IV leading to the dodecaploids from Cameroon, and (iii) contribution from nuclear clades II and IV into nuclear clade III leading to Zambian hexaploids. The organelle phylogenies suggest that polyploids might have arisen multiple times in Zambia (electronic supplementary material, figure S2), although secondary gene flow (e.g. via tetraploids) might also explain these patterns. Our results, therefore, indicate multiple admixture events between three nuclear lineages, with and without polyploidization.

## 5. Concluding remarks

We use phylogenomics of organelle and nuclear genomes to obtain a detailed picture of the phylogeographic history of the grass *A. semialata*, which is the only known species to encompass $C_3$, $C_3 + C_4$ and $C_4$ populations. The phylogenetic trees of organelle genomes, which are generally maternally inherited, indicate that seed dispersal is limited in the Central Zambezian region where *A. semialata* originated. One organelle lineage left this region via the adjacent lowlands, where warmer temperatures might have created the selective impetus for the emergence of $C_4$ photosynthesis. Seed dispersal was strongly accelerated outside of the Central Zambezian region, which likely reflects landscape differences. This led to the rapid spread of the organelle

lineage associated with the newly emerged C$_4$ trait and, to a lesser degree, of a distinct organelle lineage that migrated to southern Africa. Importantly, the patterns of nuclear genome variation indicate that pollen-mediated transport of biparentally transmitted genes occurred over longer distances and recurrently among organelle lineages. This process allowed episodic genetic exchanges between photosynthetic types, in some cases via intraspecific allopolyploidization. In particular, our study reveals an unprecedented instance of unidirectional gene flow from a C$_4$ to a non-C$_4$ genome. We conclude that pollen-mediated exchanges of nuclear genes between geographically isolated lineages allowed the rapid spread of the novel C$_4$ trait into populations inhabiting distant regions.

Data accessibility. Sequencing data were deposited in the NCBI database under BioProject PRJNA666779. Organelle genome alignments are available from the Dryad Digital Repository (https://doi.org/10.5061/dryad.zs7h44j6v) [24].

Authors' contributions. M.E.B., L.T.D., J.K.O., C.P.O. and P.-A.C. designed the study. M.E.B. did the phylogenetic analyses. L.T.D. did the allele analyses. E.V.C. did the population genomics analyses. J.K.O., S.M. and G.B. generated sequence data. O.H., R.P., S.M. and I.L. generated the genome sizes. M.R.L. did isotope analyses. M.S.V. contributed samples. M.E.B. and P.A.C. wrote the manuscript with the help of all co-authors.

Competing interests. We declare that we have no competing interests.

Funding. This study was funded by the European Research Council (grant no. ERC-2014-STG-638333), the Royal Society (grant no. RGF\EA\181050) and has benefited from 'Investissements d'Avenir' grants managed by the Agence Nationale de la Recherche (CEBA, ref. ANR-10-LABX-25-01 and TULIP, ref. ANR-10-LABX-41). Edinburgh Genomics, which contributed to the sequencing, is partly supported through core grants from the NERC (grant no. R8/H10/ 56), MRC (grant no. MR/K001744/1) and BBSRC (grant no. BB/ J004243/1). P.A.C. is funded by a Royal Society University Research Fellowship (grant no. URF\R\180022).

Acknowledgements. We thank Pauline Raimondeau for laboratory support.

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
