## [Reviewer comments · Proceedings of the Royal Society B: Biological Sciences]

Review History

RSPB-2020-1960.R0 (Original submission)

Review form: Reviewer 1

Recommendation

Accept with minor revision (please list in comments)

Scientific importance: Is the manuscript an original and important contribution to its field?

Good

General interest: Is the paper of sufficient general interest?

Excellent

Quality of the paper: Is the overall quality of the paper suitable?

Good

Is the length of the paper justified?

Yes

Should the paper be seen by a specialist statistical reviewer?

Yes

Do you have any concerns about statistical analyses in this paper? If so, please specify them explicitly in your report.

Yes

It is a condition of publication that authors make their supporting data, code and materials available - either as supplementary material or hosted in an external repository. Please rate, if applicable, the supporting data on the following criteria.

Is it accessible?

Yes

Is it clear?

Yes

Is it adequate?

Yes

Do you have any ethical concerns with this paper?

Yes

Comments to the Author

C4 photosynthesis is biologically interesting and agriculturally important, so attempts to understand evolution of this process are very important. This manuscript uses genomics and phylogeography to understand variation in a grass species that includes both C3 and C4 genotypes.

What assumptions do the phylogenetic, clustering, and admixture algorithms make regarding ploidy? (Do the likelihood methods really allow polyploidy? How does one identify orthologous loci in segmental allopolyploids?) Are these assumptions appropriate, and what might be the consequences if these assumptions are not met? (This question is very important for interpreting the validity of the authors' conclusions.)

I am not an expert on details of genome assembly or phylogenetic computation. Someone else needs to focus on those details.

What is the origin of these plant materials? Does this article include research that required permits?

Other comments:

Fig 1: The legend is incomplete. Please explain the gray and colored small and large circles located to the right of the dendrogram.

Fig 3: Additional written details are needed to interpret the figure. For example, please specify that colors follow Fig. 1, and the letter codes indicate clades as in Fig. 1.

Need to be reworded: line 58 and lines 118 - 119;

line 103: individual -> individuals

line 401: In particular, out study reveals -> In particular, our study reveals

Review form: Reviewer 2

Recommendation

Major revision is needed (please make suggestions in comments)

Scientific importance: Is the manuscript an original and important contribution to its field?

Good

General interest: Is the paper of sufficient general interest?

Excellent

Quality of the paper: Is the overall quality of the paper suitable?

Marginal

Is the length of the paper justified?

Yes

Should the paper be seen by a specialist statistical reviewer?

No

Do you have any concerns about statistical analyses in this paper? If so, please specify them explicitly in your report.

Yes

It is a condition of publication that authors make their supporting data, code and materials available - either as supplementary material or hosted in an external repository. Please rate, if applicable, the supporting data on the following criteria.

Is it accessible?

Yes

Is it clear?

Yes

Is it adequate?

Yes

Do you have any ethical concerns with this paper?

No

Comments to the Author

This manuscript by Bianconi et al. used phylogenomics and population genomics of organellar and nuclear genomes to elucidate the phylogeographic history of the grass *Alloteropsis semialata* with both C3, C3+C4 and C4 populations. They found evidence of limited seed dispersal but extensive pollen-mediated gene flow, which enabled the rapid spread of the derived C4 physiology away from its region of origin. These findings are important and, if well supported, merit to be widely known. However, I have serious concerns with the writing and the methods of data collection and analysis.

The English is rather poor, with numerous grammatical errors and/or typos (see line 55-56 as an exemplar); sometimes I feel extremely difficult to follow and understand what they are trying to say.

The method section about whole genome sequencing lacks important information about the sequencing depth for each specimen, which is necessary for readers to get a sense of the quality of

the data acquired and used in downstream analysis.

Concerning the organellar genome assembly, it is not clear to me how the authors exclude the possibility of nupts. Usually, for each position in the reference mitochondrial or plastid genome, bases were called only if the coverage was greater than a threshold specifically designed for the study according to the sequencing depth. I find nowhere in the manuscript that says about this. Without such curation of data, the plastome or mitochondrial genome sequences obtained may suffer from problems such as false positives.

For dating analysis, secondary calibration was applied; I can understand this strategy because generally no fossil records are available for grass plants, but a caveat should be made in the paper, probably both in Methods and Results, because secondary calibrations, particularly when applying a normal prior as is the case here, often results in biased or erroneous estimates. See Schenk (2016, Plos One) for potential consequences of secondary calibrations on divergence time estimates.

I also have difficulties in knowing how data was acquired about the genome composition of individuals with varied ploidy levels, especially how to distinguish between orthologs and paralogs within each polyploid genome? Perhaps this may be due to my own limitations in this area, but further clarification may be useful and would be appreciated by readers like me. In addition, do the authors have obtained evidence for both disomic and tetrasomic inheritance in the polyploidy plants when they were referred to as segmental allopolyploids?

My overall expression is that this investigation addresses a very important issue, but the writing style and methods of data acquisition and analysis as described are not sufficient, preventing me from recommending publication in its current form.

Decision letter (RSPB-2020-1960.R0)

28-Sep-2020

Dear Dr Christin:

Your manuscript has now been peer reviewed and the reviews have been assessed by an Associate Editor. The reviewers' comments (not including confidential comments to the Editor) and the comments from the Associate Editor are included at the end of this email for your reference. As you will see, the reviewers and the Editors have raised some concerns with your manuscript and we would like to invite you to revise your manuscript to address them.

When submitting your revision please upload a file under "Response to Referees" - in the "File Upload" section. This should document, point by point, how you have responded to the reviewers' and Editors' comments, and the adjustments you have made to the manuscript. We

require a copy of the manuscript with revisions made since the previous version marked as 'tracked changes' to be included in the 'response to referees' document.

Research ethics:

Use of animals and field studies:

It is a condition of publication that you make available the data and research materials supporting the results in the article. Please see our Data Sharing Policies (<https://royalsociety.org/journals/authors/author-guidelines/#data>). Datasets should be deposited in an appropriate publicly available repository and details of the associated accession number, link or DOI to the datasets must be included in the Data Accessibility section of the article (<https://royalsociety.org/journals/ethics-policies/data-sharing-mining/>). Reference(s) to datasets should also be included in the reference list of the article with DOIs (where available).

If you wish to submit your data to Dryad (<http://datadryad.org/>) and have not already done so you can submit your data via this link [http://datadryad.org/submit?journalID=RSPB&manu=\(Document not available\)](http://datadryad.org/submit?journalID=RSPB&manu=(Document%20not%20available)), which will take you to your unique entry in the Dryad repository.

Online supplementary material will also carry the title and description provided during submission, so please ensure these are accurate and informative. Note that the Royal Society will not edit or typeset supplementary material and it will be hosted as provided. Please ensure that

the supplementary material includes the paper details (authors, title, journal name, article DOI). Your article DOI will be 10.1098/rspb.[paper ID in form xxxx.xxxx e.g. 10.1098/rspb.2016.0049].

Please submit a copy of your revised paper within three weeks. If we do not hear from you within this time your manuscript will be rejected. If you are unable to meet this deadline please let us know as soon as possible, as we may be able to grant a short extension.

Best wishes,
Dr Daniel Costa
mailto:proceedingsb@royalsociety.org

Associate Editor

Comments to Author:

C4 photosynthesis is an efficient pathway, especially in the stressful warm, arid and saline environments. It is biologically interesting and important for agriculture. Although it had independently originated in terrestrial plants multiple times, it is still illusive how this complicated pathway transited from C3 photosynthesis. Using phylogenomic and population genomic approaches, authors inferred how C4 photosynthesis diverged from C3 photosynthesis in the grass *Alloteropsis semialata*. Authors suggested that photosynthesis innovation in *A. semialata* was the results of limited dispersal and isolation happened after migration to new environments. The results of this study provide the new insights to process of transition between different photosynthesis pathways. Both reviewers recognized the importance of this works. However, both of them provided insightful and critical suggestions for further improvement of this interesting manuscript. I would suggest that authors to revise this manuscript by carefully addressing all the issues pointed out by both reviewers, especially the reviewer 2, before it can be accepted for publication in PRSB.

Reviewer(s)' Comments to Author:

Referee: 1

Comments to the Author(s)

C4 photosynthesis is biologically interesting and agriculturally important, so attempts to understand evolution of this process are very important. This manuscript uses genomics and phylogeography to understand variation in a grass species that includes both C3 and C4 genotypes.

What assumptions do the phylogenetic, clustering, and admixture algorithms make regarding ploidy? (Do the likelihood methods really allow polyploidy? How does one identify orthologous loci in segmental allopolyploids?) Are these assumptions appropriate, and what might be the consequences if these assumptions are not met? (This question is very important for interpreting the validity of the authors' conclusions.)

I am not an expert on details of genome assembly or phylogenetic computation. Someone else needs to focus on those details.

What is the origin of these plant materials? Does this article include research that required permits?

Other comments:

Fig 1: The legend is incomplete. Please explain the gray and colored small and large circles located to the right of the dendrogram.

Fig 3: Additional written details are needed to interpret the figure. For example, please specify that colors follow Fig. 1, and the letter codes indicate clades as in Fig. 1.

Need to be reworded: line 58 and lines 118 - 119;

line 103: individual -> individuals

line 401: In particular, out study reveals -> In particular, our study reveals

Referee: 2

Comments to the Author(s)

This manuscript by Bianconi et al. used phylogenomics and population genomics of organellar and nuclear genomes to elucidate the phylogeographic history of the grass *Alloteropsis semialata* with both C3, C3+C4 and C4 populations. They found evidence of limited seed dispersal but extensive pollen-mediated gene flow, which enabled the rapid spread of the derived C4 physiology away from its region of origin. These findings are important and, if well supported, merit to be widely known. However, I have serious concerns with the writing and the methods of data collection and analysis.

The English is rather poor, with numerous grammatical errors and/or typos (see line 55-56 as an exemplar); sometimes I feel extremely difficult to follow and understand what they are trying to say.

The method section about whole genome sequencing lacks important information about the sequencing depth for each specimen, which is necessary for readers to get a sense of the quality of the data acquired and used in downstream analysis.

Concerning the organellar genome assembly, it is not clear to me how the authors exclude the possibility of nupts. Usually, for each position in the reference mitochondrial or plastid genome, bases were called only if the coverage was greater than a threshold specifically designed for the study according to the sequencing depth. I find nowhere in the manuscript that says about this. Without such curation of data, the plastome or mitochondrial genome sequences obtained may suffer from problems such as false positives.

For dating analysis, secondary calibration was applied; I can understand this strategy because generally no fossil records are available for grass plants, but a caveat should be made in the paper, probably both in Methods and Results, because secondary calibrations, particularly when applying a normal prior as is the case here, often results in biased or erroneous estimates. See Schenk (2016, Plos One) for potential consequences of secondary calibrations on divergence time estimates.

I also have difficulties in knowing how data was acquired about the genome composition of individuals with varied ploidy levels, especially how to distinguish between orthologs and paralogs within each polyploid genome? Perhaps this may be due to my own limitations in this area, but further clarification may be useful and would be appreciated by readers like me. In addition, do the authors have obtained evidence for both disomic and tetrasomic inheritance in the polyploidy plants when they were referred to as segmental allopolyploids?

My overall expression is that this investigation addresses a very important issue, but the writing style and methods of data acquisition and analysis as described are not sufficient, preventing me from recommending publication in its current form.

Author's Response to Decision Letter for (RSPB-2020-1960.R0)

See Appendix A.

Decision letter (RSPB-2020-1960.R1)

20-Oct-2020

Dear Dr Christin

I am pleased to inform you that your manuscript entitled "Contrasted histories of organelle and nuclear genomes underlying physiological diversification in a grass species" has been accepted for publication in Proceedings B.

Open Access

You are invited to opt for Open Access, making your freely available to all as soon as it is ready for publication under a CCBY licence. Our article processing charge for Open Access is £1700. Corresponding authors from member institutions (<http://royalsocietypublishing.org/site/librarians/allmembers.xhtml>) receive a 25% discount to these charges. For more information please visit <http://royalsocietypublishing.org/open-access>.

Paper charges

Sincerely,
Dr Daniel Costa
Editor, Proceedings B

Associate Editor:

Board Member

Comments to Author:

C4 photosynthesis is derived from ancestral C3 photosynthetic metabolism and is efficient for highly stringent environments, such as arid and saline habitats. C4 photosynthesis allows plants to expand their ecological niches and is important to crop production. Although C4 photosynthesis had independently evolved multiple times in plants, its evolutionary dynamics is difficult to be traced in the old C4 lineages. The grass *Alloteropsis semialata* is the only known species with both C4 and C3 genotypes. Therefore, it provides an ideal system to investigate how C3 photosynthetic metabolism transformed into C4 system. Authors used both the phylogenomic and population genomic approaches to infer historical demography and gene flow between different photosynthetic genotypes of *A. semialata*. Their results suggested that photosynthetic innovation happened after *A. semialata* dispersed into new environment and limited dispersal and isolated allow different genotypes to evolve. And the newly derived C4 genotypes rapidly spread through pollen gene flow. Authors had taken care the comments provided by the reviewers in this revision. Therefore, it should be suitable for publication by PRSB now.

Appendix A

Associate Editor

Comments to Author:

C4 photosynthesis is an efficient pathway, especially in the stressful warm, arid and saline environments. It is biologically interesting and important for agriculture. Although it had independently originated in terrestrial plants multiple times, it is still illusive how this complicated pathway transited from C3 photosynthesis. Using phylogenomic and population genomic approaches, authors inferred how C4 photosynthesis diverged from C3 photosynthesis in the grass *Alloteropsis semialata*. Authors suggested that photosynthesis innovation in *A. semialata* was the results of limited dispersal and isolation happened after migration to new environments. The results of this study provide the new insights to process of transition between different photosynthesis pathways. Both reviewers recognized the importance of this works. However, both of them provided insightful and critical suggestions for further improvement of this interesting manuscript. I would suggest that authors to revise this manuscript by carefully addressing all the issues pointed out by both reviewers, especially the reviewer 2, before it can be accepted for publication in PRSB.

>>> Dear Editor,

>>> We thank you for your valuable comments and those of your reviewers. We have considered all of them carefully, and provide below point-by-point responses and lists of the changes made to address each comment. In particular, we have 1) carefully improved the writing throughout, 2) added conceptual descriptions of the different methods to ensure that future readers will fully understand their purposes, and 3) redone all the organelle genome analyses with a depth filter as suggested by the second reviewer.

>>> We provide our responses in blue, preceded by three 'greater than' signs. Line numbers correspond to the revised version without track changes, but we also submit a version of the manuscript where all changes are highlighted.

>>> The different comments really improved the quality of our manuscript, and we are therefore grateful. We look forward to working with your journal.

>>> All the best,

>>> Pascal-Antoine Christin, on behalf of all the authors

Reviewer(s)' Comments to Author:

Referee: 1

Comments to the Author(s)

C4 photosynthesis is biologically interesting and agriculturally important, so attempts to understand evolution of this process are very important. This manuscript uses genomics and phylogeography to understand variation in a grass species that includes both C3 and C4 genotypes.

What assumptions do the phylogenetic, clustering, and admixture algorithms make regarding ploidy? (Do the likelihood methods really allow polyploidy? How does one identify orthologous

loci in segmental allopolyploids?) Are these assumptions appropriate, and what might be the consequences if these assumptions are not met? (This question is very important for interpreting the validity of the authors' conclusions.)

>>> Response: We agree with the reviewer that analysing polyploid data comes with its own difficulties, and it is essential to account for these during the analyses and subsequent interpretation. Below, we address the issue of ploidy for each method used in our paper, and the mitigation strategies adopted:

>>> 1) Phylogenetic analyses: these methods assume that there is a tree-like structure in the dataset. This assumption is obviously violated by the presence of reticulate evolution. We consequently used coalescence multigene species trees, which explicitly allow for different histories among genes. Because we assemble one sequence per gene per individual, this approach does however not solve the problem of different alleles of one individual having different histories. This problem can happen in diploids, but will be more marked in allopolyploids. We have computed a phylogenomic tree solely on diploid individuals and obtained the same results: (L154) - “The analysis was repeated with only confirmed diploid individuals” and (L265-268) - “The four nuclear clades previously defined within *A. semialata* [11] were retrieved with high support (Fig. 2), and similar relationships were obtained with different thresholds for missing data (Fig. S3), and when only diploid samples were included (Fig. S4)”. We conclude that the relationships among diploid individuals are not biased by the presence of polyploids.

>>> 2) Population genomics approaches (PCA and admixture): these population genetic methods do not assume a tree-like structure in the dataset, and that were used here to explore patterns of genetic variation. The tools, which complement the phylogenomic analyses, are widely used and will be familiar to many readers. However, because they are not tailored for analyses of datasets with mixed ploidy and do not incorporate ploidy level information, we do not base our conclusions on the origins of polyploids solely on these analyses. Indeed we have developed a new approach to further investigate the genome composition of these individuals using phased allele sequences.

>>> 3) The ‘genome composition’ approach: we developed this approach to circumvent potential shortcomings of the widely used methods. In this approach, individual reads representing the history of each allele are analysed independently. We can consequently confidently detect cases of reticulate evolution where distinct alleles have different histories, for example as a result of the formation of allopolyploids. Our approach solves the problem of different phylogenetic origins among alleles, for either diploids or polyploids. We now explain both the shortcomings of the two first approaches and the strength of the third one in the text (L182-186):

>>> “The multigene coalescence approach implemented above allows for different histories of loci, but not for different histories of alleles at each locus, as expected in diploid hybrids or allopolyploids. Likewise, the population genomics tools used above are not tailored for mixed ploidy datasets [39], and do not explicitly consider known ploidy differences. We consequently established the history of polyploids with phylogenetic trees of phased alleles obtained with a newly developed pipeline.”

>>> 4) Determining orthology: all gene comparisons rely on accurate orthology inference. Here we have used a stringent strategy to minimize paralogy problems. The phylogenomic analyses are based on single-copy orthologs identified using OrthoFinder, which is one of the most widely used tools for inferring homology relationships among genes. Because they combine the information of > 3,000 genes, it is unlikely that species tree analyses are biased by a very small amount of misassignment of co-orthologs. We have taken extra care to avoid paralogs in the ‘genome composition’ analyses. First, we considered only BUSCO genes, which are single-copy genes in most land plant species. Second, we performed preliminary analyses on diploid individuals that included a F1 hybrid between two divergent clades of *A. semialata*. These preliminary analyses were used to select only those markers that successfully placed one of the alleles of this F1 in the clade of its mother (clade I) and one in the clade of its father (clade IV), which would not be achieved by paralogs or other phylogenetically problematic markers. We have clarified these points by making the following edits to the text:

>>> L190-192 - “As a first step to verify that the genes were single-copy and orthologous, we considered only the 26 individuals of *A. semialata* that were diploid, including a F1 hybrid between C₃ (nuclear clade I) and C₄ (nuclear clade IV) parents (Table S2).” and L209-211 - “The 120 genes not fulfilling this criterion were discarded, as they might represent insufficiently variable genes or include *Alloteropsis*-specific paralogs.”

I am not an expert on details of genome assembly or phylogenetic computation. Someone else needs to focus on those details.

>>> Response: We have added some explanations where appropriate (especially in the ‘genome composition’ section, L181-226) and hope that this will help non-specialist readers understand the purpose of the different phylogenetic steps.

What is the origin of these plant materials? Does this article include research that required permits?

>>> Responses: The origin of each plant is indicated in Table S1. Samples obtained from herbarium collections or germplasms did not require research permits. For field collected samples, permits were acquired. Information about the research permits has been added to Table S1, in an extra column.

Other comments:

Fig 1: The legend is incomplete. Please explain the gray and colored small and large circles located to the right of the dendrogram.

>>> Response: The large circles indicate groups of accessions, and the small black circles and coloured squares indicate the ploidy levels and photosynthetic types, respectively. We have changed the symbols for ploidy levels to make them visually more explicit. The legend has been clarified (L597-601): “For each sample, the photosynthetic type is indicated with a coloured square and the ploidy level by the number of black dots. All sampled African populations are shown on the map, with circles coloured based on the group indicated on the right of the phylogeny. Arrows indicate putative dispersal events.”

Fig 3: Additional written details are needed to interpret the figure. For example, please specify that colors follow Fig. 1, and the letter codes indicate clades as in Fig. 1.

>>> Response: This has been specified (L611-613): “The proportion of alleles of single-copy exons assigned to each of the four nuclear clades of *A. semialata*, numbered and coloured as in Fig. 2, are represented by colours in bars (see Table S2).”

Need to be reworded: line 58 and lines 118 - 119;

>>> Response: These two sentences have been corrected.

line 103: individual -> individuals

>>> Response: Corrected.

line 401: In particular, our study reveals -> In particular, our study reveals

>>> Response: Corrected.

Referee: 2

Comments to the Author(s)

This manuscript by Bianconi et al. used phylogenomics and population genomics of organellar and nuclear genomes to elucidate the phylogeographic history of the grass *Alloteropsis semialata* with both C3, C3+C4 and C4 populations. They found evidence of limited seed dispersal but extensive pollen-mediated gene flow, which enabled the rapid spread of the derived C4 physiology away from its region of origin. These findings are important and, if well supported, merit to be widely known. However, I have serious concerns with the writing and the methods of data collection and analysis.

The English is rather poor, with numerous grammatical errors and/or typos (see line 55-56 as an exemplar); sometimes I feel extremely difficult to follow and understand what they are trying to say.

>>> Response: We have carefully checked the text and removed all typos and grammatical mistakes. We have made numerous changes to improve the text and keep its length below the journal limit. These can be seen in the version with tracked changes.

The method section about whole genome sequencing lacks important information about the sequencing depth for each specimen, which is necessary for readers to get a sense of the quality of the data acquired and used in downstream analysis.

>>> Response: The sequencing depth is given in Table S1. We now explicitly refer to it in the methods, and give some statistics (L87-88): “The expected sequencing depth ranged from 0.6 to $58.7 \times$ (median = $4.9 \times$; Table S1).”. In addition, we specify the depth of the subset used for the ‘genome composition’ analyses (L192-194 - “These individuals were sequenced as 250 bp paired-

end reads (insert size = 550 bp) with an expected sequencing depth between 4.7 and 58.8”).

Concerning the organellar genome assembly, it is not clear to me how the authors exclude the possibility of nupts. Usually, for each position in the reference mitochondrial or plastid genome, bases were called only if the coverage was greater than a threshold specifically designed for the study according to the sequencing depth. I find nowhere in the manuscript that says about this. Without such curation of data, the plastome or mitochondrial genome sequences obtained may suffer from problems such as false positives.

>>> Response: We had originally used a depth filter only for the mitochondrial genome analysis, which we agree was not optimal. We have now reassembled all plastid and mitochondrial genome sequences with a consistent depth filter that took into account the expected sequencing depth (i.e. nuclear genome expected sequencing depth; see Table S1) for each accession. We have redone all the subsequent analyses and updated all the figures. Because we already considered only reads mapped uniquely to the reference organelles (and not the nuclear genome, also included in the reference), the extra depth filter ensures that the possibility of nupts is excluded. This is now specified in the text (L116-120):

>>> “Variant sites from the reads uniquely mapped to each organelle were incorporated into a majority consensus sequence using the mpileup function of Samtools v1.9 [20] implemented in a bash-scripted pipeline [12]. Only sites covered by more than five times the expected sequencing depth of the nuclear genome (Table S1) were called, discarding potential organelle-nuclear transfers.”

For dating analysis, secondary calibration was applied; I can understand this strategy because generally no fossil records are available for grass plants, but a caveat should be made in the paper, probably both in Methods and Results, because secondary calibrations, particularly when applying a normal prior as is the case here, often results in biased or erroneous estimates. See Schenk (2016, Plos One) for potential consequences of secondary calibrations on divergence time estimates.

>>> Response: We do agree with the reviewer that secondary calibrations are imperfect, but as they state, they are the only option for groups without fossil records (the fossil record is sparse for grasses in general and non-existent for *Alloteropsis*). Despite these caveats, secondary calibrations are expected to produce accurate relative ages among lineages (as would be obtained by fixing the root to an arbitrary value) and the use of the secondary calibration points provide indications of the absolute ages. We have added a statement in the methods, with reference to the suggested paper (L131-133): “While secondary calibrations are imperfect and can result in younger estimates and underestimated uncertainties [25], they are the only option in the absence of fossils of *Alloteropsis* and provide accurate relative ages and indicative absolute ages.” We also acknowledge the caveat in the Results (L244-245): “The ages estimated on the mitochondrial and plastid genomes were overall similar, and discrepancies can result from the use of secondary calibrations.”

I also have difficulties in knowing how data was acquired about the genome composition of individuals with varied ploidy levels, especially how to distinguish between orthologs and paralogs within each polyploid genome? Perhaps this may be due to my own limitations in this area, but further clarification may be useful and would be appreciated by readers like me. In addition, do the

authors have obtained evidence for both disomic and tetrasomic inheritance in the polyploidy plants when they were referred to as segmental allopolyploids?

>>> Response: We have explained in response to Reviewer 1 how we minimized the risk of paralogy problem, but think the question of this reviewer refers exclusively to the ‘genome composition’ analysis. In short, alleles of allopolyploids are not orthologs in the strict sense. Here, we identify good orthologs from diploid genomes and then extract homologs from the polyploid genomes. For polyploids, we then perform gene trees for each pair of reads, which allows establishing their origin. We have clarified this in the methods:

>>> L185-186 - “We consequently established the history of polyploids with phylogenetic trees of phased alleles obtained with a newly developed pipeline.”

>>> L186-188 - “o reduce risks of paralogy, we considered only near-universally single-copy orthologs in land plants according to BUSCO v3.0.2 [40].”

>>> L190-192 - “As a first step to verify that the genes were single-copy and orthologous, we considered only the 26 individuals of *A. semialata* that were diploid, including a F1 hybrid between C₃ (nuclear clade I) and C₄ (nuclear clade IV) parents (Table S2).”

>>> L209-212 - “The 120 genes not fulfilling this criterion were discarded, as they might represent insufficiently variable genes or include *Alloteropsis*-specific paralogs. The remaining 180 phylogenetically-informative genes were retained for downstream analysis.”

>>> L213-215 - “Because phasing is difficult with polyploids, we incorporated each pair of reads from the polyploids into the phylogenetic tree containing the diploid phased alleles and assigned them to the clade in which they were positioned.”

>>> Regarding the segmental allopolyploidy question, we have not verified the inheritance pattern in the present study. It has however been shown before that hexaploids from South Africa form mainly bivalents, with some multivalents (up to hexavalents – Liebenberg and Fossey 2001). We have added this information to the introduction. Because we agree that ‘segmental allopolyploidy’ has a precise meaning linked to the formation of multivalents and because this has not been verified for all polyploid lineages of *Alloteropsis semialata*, we have changed it to ‘intraspecific allopolyploidy’ in the introduction and the conclusions. The sentence from the introduction now reads as follows (L59-61): “In addition, previously reported discrepancies between mitochondrial and plastid genomes [12] might reflect the footprint of intraspecific allopolyploidization, as previously suggested based on cytological analyses [13].”

My overall expression is that this investigation addresses a very important issue, but the writing style and methods of data acquisition and analysis as described are not sufficient, preventing me from recommending publication in its current form.

>>> Response: We are glad that the reviewer appreciates the importance of the study and hope that our extensive edits of the text and additions to the description of methods, including newly assembled organelle genomes based on the reviewer suggestions, will help clarify our message.